# The Clinical Impact of Metagenomic Next-Generation Sequencing (mNGS) Test in Hospitalized Patients with Suspected Sepsis: A Multicenter Prospective Study

**DOI:** 10.3390/diagnostics13020323

**Published:** 2023-01-16

**Authors:** Yi-Hui Zuo, Yi-Xing Wu, Wei-Ping Hu, Yan Chen, Yu-Ping Li, Zhen-Ju Song, Zhe Luo, Min-Jie Ju, Min-Hua Shi, Shu-Yun Xu, Hua Zhou, Xiang Li, Zhi-Jun Jie, Xue-Dong Liu, Jing Zhang

**Affiliations:** 1Department of Pulmonary and Critical Care Medicine, Zhongshan Hospital, Shanghai Medical College, Fudan University, Fenglin Road 180, Xuhui District, Shanghai 200032, China; 2Department of Pulmonary and Critical Care Medicine, Second Xiangya Hospital of Central South University, Mid Renmin Road 139, Changsha 410011, China; 3Department of Pulmonary and Critical Care Medicine, The First Affiliated Hospital of Wenzhou Medical University, Fuxue Alley 2, Lucheng District, Wenzhou 325015, China; 4Department of Emergency, Zhongshan Hospital, Shanghai Medical College, Fenglin Road 180, Xuhui District, Shanghai 200032, China; 5Department of Critical Care Medicine, Zhongshan Hospital, Shanghai Medical College, Fudan University, Fenglin Road 180, Xuhui District, Shanghai 200032, China; 6Department of Pulmonary Medicine, The Second Affiliated Hospital of Suzhou University, Sanxiang Road 1055, Gusu District, Suzhou 215004, China; 7Department of Pulmonary and Critical Care Medicine, Tongji Hospital of Tongji Medical College, Huazhong University of Science and Technology, Jiefang Avenue 1095, Qiaokou District, Wuhan 430030, China; 8Department of Pulmonary and Critical Care Medicine, The First Affiliated Hospital of Zhejiang University, Bunichi West Road 1367, Yuhang District, Hangzhou 310003, China; 9Department of Critical Care Medicine, Central Hospital of Minhang District, Shanghai Medical College, Fudan University, Xinsong Road 170, Minhang District, Shanghai 201199, China; 10Department of Pulmonary and Critical Care Medicine, Shanghai Fifth People’s Hospital, Shanghai Medical College, Fudan University, Ruili Road 128, Minhang District, Shanghai 200240, China; 11Department of Pulmonary and Critical Care Medicine, Qingdao Municipal Hospital, Jiaozhou Road 1, North City District, Qingdao 266071, China

**Keywords:** sepsis, metagenomic next generation sequencing (mNGS), etiological diagnosis, antibiotic therapy, prognosis

## Abstract

Background: Metagenomic Next Generation Sequencing (mNGS) has the potential to detect pathogens rapidly. We aimed to assess the diagnostic performance of mNGS in hospitalized patients with suspected sepsis and evaluate its role in guiding antimicrobial therapy. Methods: A multicenter, prospective cohort study was performed. We enrolled patients with suspected sepsis, collected clinical characteristics and blood samples, and recorded the 30-day survival. Diagnostic efficacy of mNGS test and blood culture was compared, and the clinical impact of mNGS on antibiotic regimen modification was analyzed. Results: A total of 277 patients were enrolled, and 162 were diagnosed with sepsis. The mortality was 44.8% (121/270). The mNGS test exhibited shorter turn-out time (27.0 (26.0, 29.0) vs. 96.0 (72.0, 140.3) hours, *p* < 0.001) and higher sensitivity (90.5% vs. 36.0%, *p* < 0.001) compared with blood culture, especially for fungal infections. The mNGS test showed better performance for patients with mild symptoms, prior antibiotic use, and early stage of infection than blood culture, and was capable of guiding antibiotic regimen modification and improving prognosis. Higher reads of pathogens detected by mNGS were related to 30-day mortality (*p* = 0.002). Conclusions: Blood mNGS testing might be helpful for early etiological diagnosis of patients with suspected sepsis, guiding the antibiotic regimen modification and improving prognosis.

## 1. Introduction

Infections are common causes of hospitalization, which could progress into sepsis. According to the latest definition of sepsis in 2016, sepsis is a disorder of the host’s inflammatory response to infection, resulting in homeostasis imbalance and life-threatening organ dysfunction syndrome. The clinical diagnostic criteria include Systemic Inflammatory Response Syndrome (SIRS), the quick Sequential (Sepsis-related) Organ Failure Assessment (qSOFA) score of ≥2 points, and high potentially fatal risk [1].

In recent years, the mortality of sepsis has not decreased in spite of the development of novel antibiotics and updated treatments [2]. Sepsis still remains the primary cause of ICU admission and death. The high mortality is partly caused by missing optimal timing for treatment when sepsis is clinically diagnosed, making it a serious threat to public health and life [1,3]. For one thing, the effective indicators for early diagnosis of sepsis are still lacking. For another thing, the lack of rapid identification of pathogens hinders the timely targeted treatment for patients with infectious diseases [3,4,5,6]. The main clinical techniques for the etiological diagnosis of sepsis include smears and cultures of clinical specimens such as blood. As the gold standard for diagnosing bloodstream infections, blood culture has obvious shortcomings, such as being time-consuming and low sensitivity.

Based on the above, it is necessary to explore more sensitive and specific diagnostic tests for patients with infections and sepsis. High-throughput Next Generation Sequencing (NGS) is a newly developed technology of nucleic acid detection in recent years. The most prominent advantage of metagenomic NGS (mNGS) is the rapid and accurate detection of pathogens that dramatically reduces the time required for clinical diagnosis, which only takes about 30 h [7,8]. Besides, the diagnostic efficacy of mNGS will not be impaired by the application of antibiotics [9]. Some studies also demonstrated that the NGS test could detect antimicrobial resistance (AMR) genes [10,11] and may help clinicians adjust the antibiotic regimen. Apart from blood samples, the mNGS test is applicable to multiple kinds of samples, including resected heart valve [12], cerebrospinal fluid [13], vitreous samples [14], and formalin-fixed and paraffin-embedded (FFPE) tissue specimens [15].

However, the role of mNGS in the management of patients with potential risks of bloodstream infections is still controversial in terms of its accuracy and guidance on optimal antimicrobial use. Therefore, we performed a prospective multicenter study with the following purposes: (1) to assess the value of the mNGS test in the pathogenic diagnosis of suspected bloodstream infection and explore whether the mNGS test could improve clinical prognosis or guide antibiotic use; (2) to identify the features of patients who might benefit from mNGS.

## 2. Materials and Methods

### 2.1. Study Design and Population

A multicenter, prospective cohort study was performed. The participating hospitals and number of cases are listed in Appendix A. Zhongshan Hospital of Fudan University was the central institution responsible for this study. Sample size was determined by the principal investigator based on clinical research experience and hospital capacity.

The inclusion criteria were as follows: (1) Age > 18 years and ≤90 years old. (2) Patients who met any 1 of the following 3 criteria: ① respiratory rate (per minute) ≥ 22; ② systolic blood pressure ≤ 100 mmHg; ③ altered mental status. (3) Patients suspected of having any of the following conditions: ①bloodstream infection; ② community-acquired pneumonia (CAP); ③ hospital-acquired pneumonia (HAP); ④ peritonitis; ⑤ acute purulent bile duct inflammation; ⑥ acute pyelonephritis; ⑦ skin and soft tissue infection. (4) Signature of informed consent.

The exclusion criteria were the existence of any of the following: (1) age < 18 years or >90 years old; (2) unable to cooperate with researchers.

### 2.2. Ethics Approval

The ethics committee of Zhongshan Hospital approved this study (Number: B2018-182R). All the participants were fully informed of the details before recruitment and provided their written informed consent. The study was registered with the Chinese Clinical Trial Registry (Number: ChiCTR1800019187).

### 2.3. Data Collection

Demographic statistics and clinical characteristics were collected after signing the informed consent. Demographic statistics included age, gender, height, and weight. Clinical characteristics were as follows: (1) previous and present medical history; (2) assessment of current disease severity, including the Acute Physiologic and Chronic Health Evaluation (APACHE) II score, SOFA and qSOFA score; (3) the number of days from the infection onset to enrollment in the study where infection onset was defined as the time when the patient had a fever and elevated leukocyte; (4) laboratory examination on the day of enrollment, including blood routine, leukocyte count, PCT; (5) results of blood culture; (6) records of antibiotic regimen within 30 days after enrollment, and modification based on pathogen tests. The modification of antibiotic regimen was defined as any change of antibiotic agents within 2–7 days after enrollment, causing the results of mNGS test and blood culture during this period; (7) survival status at day 30 of follow-up.

### 2.4. mNGS Detection

Blood samples were collected at enrollment and stored at −80 °C. Within 24 h, the blood samples were transferred to BGI Medical Laboratory Institute (Wuhan, China) for mNGS detection. The mNGS detection was conducted according to the manufacturer’s protocol, which is reported in detail in Appendix A.

With reference to previous literature, we have formulated the following criteria for positive mNGS results [16].

For bacteria (mycobacteria excluded), fungi, viruses, and parasites: mNGS identified a microbe (genus level) as positive whose reads were 3-fold greater than that of any other microbes. If only one pathogen was detected, it would be regarded as positive.

For mycobacteria: Mycobacterium tuberculosis was considered positive as long as there was more than 1 read mapped to the species or genus level.

Under the condition that the above criteria were met, the patient’s clinical characteristics were taken into consideration. Parvovirus, human herpes virus 5, etc., were appropriately excluded by clinicians and laboratory staff due to the consideration of laboratory contamination. Other results were treated as positive.

### 2.5. Blood Culture

The blood culture was carried out in the microbiological laboratory of each participating hospital. Blood samples were collected from the bilateral upper limbs of patients on the day of enrollment and cultured as the standard procedure. Positive results of bilateral samples were defined as positive blood culture, and positive results of unilateral samples were considered contamination.

At the end of the follow-up, the final etiological conclusion of each patient was made by attending physicians and the principal investigator in each hospital based on microbiological results, clinical features, and response to the treatment. Controversial cases were discussed in a seminar of all investigators from different centers, and a final diagnosis was determined.

The final etiological conclusion was taken as the culprit pathogen, and the reads of the culprit pathogen in mNGS reports were applied in the statistical analysis.

### 2.6. Determination of Inflammatory Factors

Plasma samples of mNGS-positive patients were taken for inflammatory factor detection using the Human Cytokine Screening 48-Plex Services Kit (BIORAD, Catalog number: 12007283. Hercules, CA, USA) according to the manufacturer’s instructions.

### 2.7. Statistical Analysis

The analysis was performed with SPSS22.0 (SPSS Inc., Chicago, IL, USA) and GraphPad Prism 8.0.1 (GraphPad Software Inc., San Diego, CA, USA).

Patients were divided into different subgroups according to clinical parameters. Continuous variables with normal distribution were represented as means ± standard deviation, and the others were represented as medians (1st quartile, 3rd quartile). Categorical variables were shown as *n* (%) and analyzed by Chi-square analysis. After the test for normality with a one-sample Kolmogorov–Smirnoff test, continuous variables that conform to normal distribution were compared by the *t*-test, and the non-normally distributed variables were compared by the Mann–Whitney *U* test. The same statistical method was applied when comparing reads of pathogens in different subgroups. The McNemar test was used to compare the sensitivity and specificity of the mNGS test and blood culture. Receiver Operating Characteristic (ROC) curves were analyzed, and the value of the Area Under Curve (AUC) was calculated. The correlations between reads of pathogens and pro-inflammatory factors were analyzed by univariate linear regression. *p* < 0.05 was defined as statistical significance.

## 3. Results

A total of 285 patients were screened, and 277 patients were enrolled and included in the final statistical analysis.

### 3.1. Demographic and Basic Clinical Information

There were 195 males and 82 females, with an average age of 62 (49, 70) years old. According to the definition of sepsis in 2016 (qSOFA score of ≥2 points) [1], 162 patients were diagnosed with sepsis, and 115 patients without sepsis. The mortality at the 30-day follow-up was 44.81% (121/270). Table 1 presents the basic patient information.

### 3.2. Characteristics of Infection

The average time from symptom onset to enrollment was 8 (3, 17) days, with leukocyte counts of 10.16 (6.70, 15.69) (×10^9^/L) and neutrophil counts of 9.03 (5.39, 13.38) (×10^9^/L). Detailed data are shown in Table 2.

Figure 1 shows the etiological results. Among the 148 patients with confirmed pathogens, bacterial infections were the most common (118 cases). *Klebsiella pneumoniae* (22 cases) counted as the first leading pathogen, followed by *Acinetobacter baumannii* (19 cases) and *Pseudomonas aeruginosa* (11 cases). For patients with bacterial infection, 75 were diagnosed with sepsis, and the positive rates of blood culture and mNGS test were 36.0% (27/75) and 86.67% (65/75), respectively. Forty-three of them were diagnosed as non-sepsis, while the positive rates of blood culture and mNGS test were 39.53% (17/43) and 90.70% (39/43), respectively. The overall positive rates of blood culture and mNGS test in bacterial-infected patients were 37.29% and 88.14%, respectively. *Adenovirus* (3 cases) was the most common pathogen in viral infections (8 cases), which were all detected by mNGS. Four cases were diagnosed with *Mycobacterium tuberculosis* infection while their blood culture results were negative, and only two of them had positive mNGS results.

Of the patients with fungi infections (22 cases), 11 of them had histories of organ transplant or malignant tumors, and the rest were suffering from long-term pneumonia or severe cardiac dysfunction. Suspected pathogens were *Pneumocystis* (13 cases), *Aspergillus* (7 cases), and *Candida* (2 cases). All of them were infected in the lungs. Twelve of the fungi-infected patients were diagnosed with sepsis, while the rest (10) were non-sepsis. The positive rates of the mNGS test were both 100.0% in fungal infections, while only 1 of the 22 patients had a positive blood culture result of *Candida*.

### 3.3. Performance of mNGS Test in Comparison with Culture

A total of 217 patients had blood cultures, of which 50 were positive and 167 were negative. Among the 277 patients’ mNGS test results, 140 were positive, 135 were negative, and 2 were missing.

The percentage of patients who were identified with the culprit pathogen increased from 16.25% (45/277) with culture to 52.71% (146/277) when combining mNGS and culture. According to the final etiological conclusion, the sensitivity, specificity, positive predictive value (PPV), and negative predictive value (NPV) of blood culture were 36.00% (45/112), 94.57% (87/105), 90.00% (45/50), and 52.10% (87/167), respectively, while the results of mNGS tests were 90.54% (134/148), 94.53% (121/128), 95.71% (134/140), and 89.63% (121/135), respectively. The AUC values of the blood culture and mNGS test were also calculated, which were 0.653 and 0.929 (*p* < 0.001), respectively.

Besides, the average turn-out time required for mNGS tests in our study was 27.0 (26.0, 29.0) hours, which was significantly shorter than the average time required for blood culture (96.0 (72.0, 140.3) hours (*p* < 0.001)).

### 3.4. Comparison of mNGS Test and Blood Culture in Different Subgroups

We divided the patients into different subgroups according to various factors and analyzed the diagnostic efficacy of the mNGS test and blood culture in each subgroup. As shown in Appendix A, the sensitivity of mNGS in each subgroup significantly preceded blood culture, while the specificity remained not inferior to blood culture.

Although there was no statistical significance, the sensitivity of blood culture exhibited a downward trend in subgroups of patients within 5 days of infection onset, patients without fever, and patients who applied antibiotics before enrollment.

### 3.5. Reads of Responsible Pathogens in mNGS Reports Were Associated with the Prognosis of Patients and the Level of Inflammatory Factors

Among the mNGS-positive patients, we compared the reads of responsible pathogens in mNGS reports in patients with different prognoses (shown in Figure 2). The results showed that the reads in the non-survival group were higher than that in the survival group, and this trend remained significant between non-survival and survival patients in the sepsis group.

We detected the levels of inflammatory factors in patients with positive mNGS results. We confirmed the positive correlations between the plasma levels of interleukin 1-β (IL-1β), eotaxin, interferon-α (IFN-α), tumor necrosis factor-α (TNF-α), and the reads of pathogens (Appendix A), and the last three remained significant in sepsis patients (Appendix A). The correlation between specific cytokine profile and pathogens was studied. We enrolled *Klebsiella pneumoniae, Acinetobacter baumannii,* and *Pseudomonas aeruginosa* into the analysis, and no statistical significance was found.

### 3.6. Microbial Detection Facilitated the Modification of Antimicrobial Prescription and Improved the Prognosis of Patients

Of all the patients, 151′s antibiotic regimens were modified within 2–7 days after the enrollment, and the rest remained with the original antibiotic regimen. Furthermore, 53 patients de-escalated antibiotic application, and 98 upgraded their antibiotic regimen. We observed that patients who modified antibiotic regimens had higher 30-day survival (63.3% vs. 45.5%, *p* = 0.005), a trend also obtained in the sepsis group (58.5% vs. 37.7%, *p* = 0.009).

Among the mNGS-negative patients (135 cases), 25 patients de-escalated antibiotic use with a 30-day mortality of 32.0%, while the other 110 patients remained on the original antibiotic regimens with a 30-day mortality of 40.0%. No statistical significance was found between the two groups.

## 4. Discussion

Our results demonstrate that mNGS took less time compared with traditional blood culture and was able to diagnose infectious diseases with high efficacy, especially in patients suspected of fungal infection, patients with mild symptoms, prior antibiotic application, or early-stage infection. Higher reads of pathogens were related to higher 30-day mortality and sepsis state, as well as higher levels of inflammatory factors, indicating that the reads of pathogens reflected the pathogen load and might be related to severity and prognosis. In addition, we found that patients whose antibiotic regimens were modified based on microbial tests had a higher 30-day survival rate, while the negative mNGS results might help with the de-escalation of antibiotics without worsening the prognosis. Our results confirmed the clinical significance of the mNGS test.

Blood culture is the gold standard for pathogen detection in bloodstream infections. However, the sensitivity of blood culture fluctuates with the severity and stage of infection [8,16,17,18], which was reported as 30–60% [19]. In our study, blood culture had high specificity compared with mNGS, while the sensitivity was only 36%.

Long Y et al. investigated the diagnostic efficiency of mNGS in ICU patients. They reported that the sensitivity of mNGS was 30.77% (24/78), which was significantly higher than 12.82% (10/78) using blood culture [20]. In our study, blood mNGS increased the etiological diagnosis rate from 16.25% to 52.71% in hospitalized patients with possible bloodstream infection.

A previous study showed that the time to positivity was 12–48 h for blood culture, and pathogen identification often took 5–7 days in the clinical setting [21]. In the current study, the average report time of mNGS was 27.0 (26.0, 29.0) h. Compared with blood culture, mNGS not only exhibited quick detection speed, but also could detect multiple pathogens, including bacteria, fungi, and viruses. The detection of culture-negative pathogens was one of mNGS’s most significant advantages [22].

In our study, the performance of mNGS was optimal in various conditions, especially those with mild symptoms, prior antibiotic treatment, or early stage of disease. This is consistent with Grumaz et al.’s study, which compared the blood culture and mNGS test results of patients with septic shock in different periods [9]. With the application of antibiotics and the improvement of the patient’s condition, the positive rate of blood culture gradually decreased, while mNGS was not affected, and the sensitivity remained higher than that of blood culture [9].

Higher reads of pathogens in mNGS reports were related to higher plasma levels of pro-inflammatory factors, indicating stronger inflammation. It indirectly verified that reads of pathogens in mNGS reports could reflect the pathogen load. Besides, patients who died within 30 days possessed higher reads of pathogens. The correlation reminded us that more attention should be paid when handling patients with high reads of pathogens in clinic. The earlier the intervention is taken, the better the clinical prognosis [23].

We explored the clinical impact of mNGS on prognosis. Patients whose antibiotic regimens were modified based on pathogenic test had significantly higher 30-day survival. However, Grumaz et al. reported different results [9]. In their study, 24 patients with septic shock changed antibiotic regimen based on mNGS results, while 17 patients followed the original regimen. The mortality of the former within 28 and 90 days was higher than that of the latter (25% vs. 11.8%, *p* = 0.261; 37.5% vs. 23.5%, *p* = 0.285, respectively) [9]. We speculated that the difference might be attributed to different sample sizes and patient populations. The overall mortality in our research was high (44.8%), which made the prognosis improvement more evident. Furthermore, the clinical benefit obtained from the pathogenic test should be credited to the combination of mNGS and blood culture. Whether the mNGS test can play a positive role in ameliorating the prognosis of infected patients and how needs to be further explored. In addition, we innovatively found that negative mNGS results could facilitate antibiotic de-escalation without worsening survival. This might be beneficial to the appropriate use of antibiotics.

Some studies reported that the NGS test was able to detect antimicrobial resistance (AMR) gene [10,11], and might help clinicians choose sensitive antibiotics. We tried to detect AMR genes using mNGS in the study, but the results were poorly correlated with culture (data not shown). For the current NGS technology used clinically, challenges remain in detecting AMR genes. Firstly, only through whole-genome sequencing can we acquire accurate results of AMR genes. Secondly, we were unable to determine which microorganism the AMR genes belonged to through plasma detection unless the pathogen strain was provided. Thirdly, even if an AMR gene was detected, the corresponding resistance phenotype might not exist since AMR genes can be silenced, inactivated, or poorly expressed. A combination of RNA+DNA mNGS may achieve high clinical efficacy [24].

Our study had several limitations. Firstly, we carried out blood culture and mNGS tests on the day of enrollment but failed to monitor the changes in pathogen detection rate during the entire disease process. Secondly, the changes in cytokines were not monitored. Thirdly, we only analyzed the results of blood samples. Therefore, the dynamic monitoring of clinical manifestation and microbial tests should be considered in future research.

The major superiority of our study was the prospective multicenter design that guaranteed the diversity of patients and reduced the possibility of data bias. Moreover, we included the plasma levels of pro-inflammatory factors into our research and explored the relationship between the reads of mNGS tests and pro-inflammatory factors, which have been rarely studied in previous research.

## 5. Conclusions

The mNGS technology was useful in the early stage of possible bloodstream infections, especially in patients with mild symptoms and a history of antibiotics use, or patients in the early stage of infection. It might contribute to the modification of antibiotic regimen and further improve the prognosis. The reads of pathogens in mNGS reports could reflect the patient’s pathogen load, as well as the severity of sepsis and the prognosis of patients. mNGS is expected to complement traditional detection methods in selected patients.

## Registration

The study was registered on the Chinese Clinical Trial Registry (Number: ChiCTR1800019187) on 30 October 2018 (Retrospectively registered). URL: http://www.chictr.org.cn/edit.aspx?pid=30239&htm=4 (accessed on 24 January 2019).

## Figures and Tables

**Figure 1 diagnostics-13-00323-f001:**
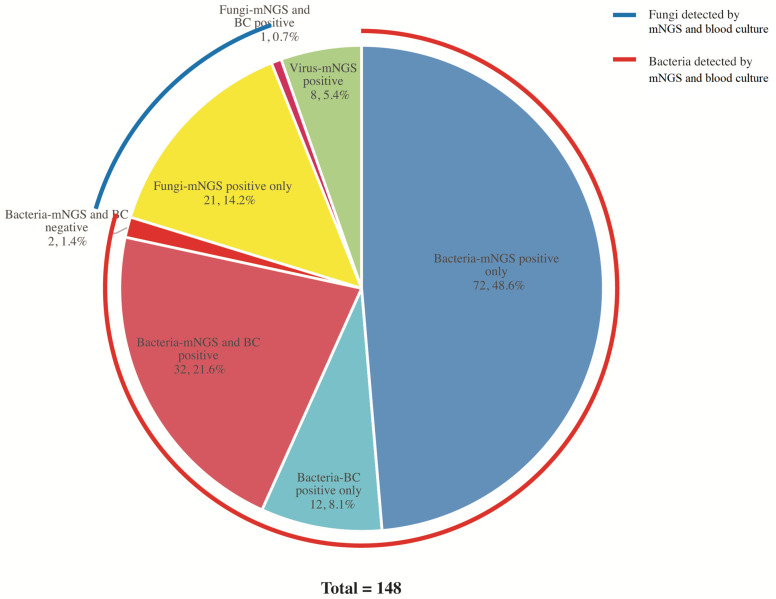
Pathogens detected by mNGS and Blood Culture (BC).

**Figure 2 diagnostics-13-00323-f002:**
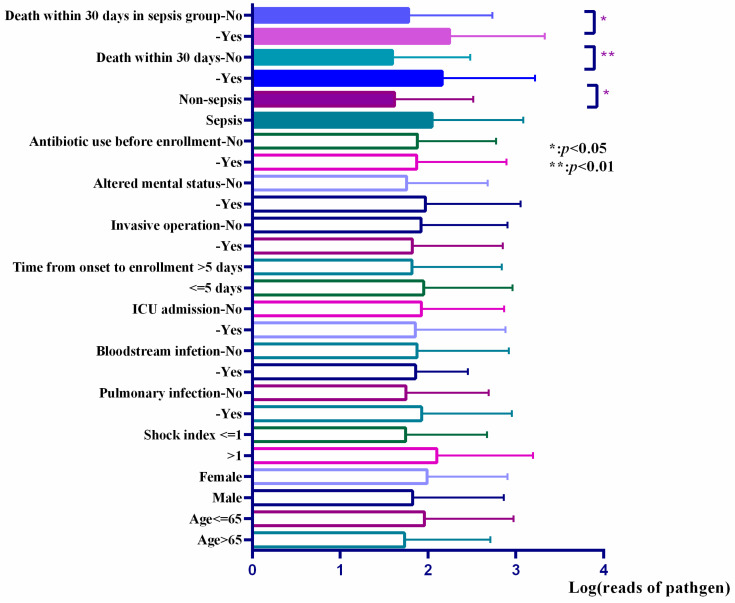
Reads of pathogens in different subgroups.

**Table 1 diagnostics-13-00323-t001:** Demographic and basic clinical characteristics of hospitalized patients with and without sepsis.

Basic Information	Sepsis Group (*n* = 162)	Non-Sepsis Group (*n* = 115)	*p*-Value
Gender, male/female	114/48	81/34	0.991
Age, median (q1, q3)	63 (50, 69)	60 (43, 72)	0.451
Comorbidities (*n* (%))			
Pulmonary disease *^a^*	13 (8.02%)	12 (10.43%)	0.490
Congestive heart failure	11 (6.79%)	4 (3.48%)	0.230
Cerebrovascular disease	19 (11.73%)	11 (9.57%)	0.568
Diabetes	34 (20.99%)	14 (12.17%)	0.056
Hepatic cirrhosis	3 (1.85%)	0 (0%)	0.269
Acute/chronic renal failure	23 (14.20%)	15 (13.04%)	0.783
Smoking history (*n* (%))	56 (34.57)	32 (27.83%)	0.235
Antibiotic use before enrollment *^b^* (*n* (%))	144 (88.89%)	104 (92.04%)	0.679
Invasive procedures before onset of symptoms *^c^* (*n* (%))	92 (56.79%)	55 (47.83%)	0.141
Corticosteroids/immunosuppressive drug/cytotoxic chemotherapy before onset (*n* (%))	38 (23.46%)	48 (41.74%)	0.001
Recent surgery/trauma history *^d^* (*n* (%))	54 (33.33%)	39 (33.91%)	0.920
ICU admission (*n* (%)])	141 (87.04%)	71 (61.74%)	<0.001

*^a^*: including chronic obstructive pulmonary disease (COPD), asthma, interstitial lung disease, structural lung disease; *^b^*: antibiotics application within 2 weeks before enrollment; *^c^*: invasive procedures including punctures and drainages, tracheal intubation, urinary catheterization, arteriovenous catheterization, and superficial vein indwelling needle puncturing; *^d^*: surgery/trauma history within 3 months before enrollment. Abbreviations: ICU, intensive care unit.

**Table 2 diagnostics-13-00323-t002:** The infection characteristics of patients in sepsis and non-sepsis groups.

Infection Characteristics	Sepsis Group (*n* = 162)	Non-Sepsis Group (*n* = 115)	*p*-Value
Time from symptom onset to enrollment (d), median (q1, q3)	5.5 (2, 15)	11 (5, 22)	0.008
Site of infection (*n* (%))			
Pulmonary infection	114 (80.28%)	70 (60.87%)	0.100
Extrapulmonary infection	18 (11.11%)	8 (6.96%)	0.247
APACHE II (means ± SD)	19.97 ± 8.33	14.87 ± 8.28	<0.001
Shock index (means ± SD)	1.04 ± 0.36	0.81 ± 0.19	<0.001
Fever (*n* (%))	117 (72.22%)	82 (71.30%)	0.867
Altered mental status *^a^* (*n* (%))	115 (70.99%)	20 (17.39%)	<0.001
Death within 30 days (*n* (%))	82, 51.57%	39, 35.14%	0.008
PCT (ug/L) [median (q1, q3)]	1.92 (0.36, 11.17)	0.53 (0.22, 3.82)	0.214
WBC count (×10^9^/L) (median (q1, q3))	10.83 (6.93, 15.65)	9.79 (6.10, 15.76)	0.410
Neutrophil count (×10^9^/L) (median (q1, q3))	9.07 (5.73, 13.42)	8.84 (5.00, 13.03)	0.475

*^a^*: including coma, delirium, ambiguity of consciousness, et al. Abbreviations: APACHE II, the Acute Physiologic and Chronic Health Evaluation II score; PCT, procalcitonin; WBC, white blood cell.

## Data Availability

The datasets used and/or analyzed during the current study are available from the corresponding author on reasonable request.

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
