# Peer review of "The Clinical Impact of Metagenomic Next-Generation Sequencing (mNGS) Test in Hospitalized Patients with Suspected Sepsis: A Multicenter Prospective Study"

_diagnostics, 2023, doi:10.3390/diagnostics13020323_

Round 1

Reviewer 1 Report

In the submitted study, the authors aimed to assess the diagnostic performance of mNGS (Metagenomic Next Generation Sequencing) in hospitalized patients with suspected sepsis and evaluate its role in guiding antimicrobial therapy. They performed a multi-center, prospective cohort study. Researchers enrolled patients with suspected sepsis, collected clinical characteristics and blood samples and recorded the 30-day survival. The diagnostic efficacy of mNGS test and blood culture was compared, and the clinical impact of mNGS on antibiotic regimen modification was analyzed.  Overall the study is thorough and well-researched.  There are a few minor concerns that the authors should address.

  1. Authors should calrify the significance of R values in table 5.
  2. Authors should recheck the references- 7, 11, 18.  
  3. Its interesting that authors ahs mentioned the limitations of this study. Authors should mention how other researchers can learn from these limitations.
  4. There are multiple studies done for sepsis and anti-microbial therapy. Authors should mention and discuss them in brief in the introduction section. 

Author Response

Reply to reviewer 1

We appreciate your extremely constructive and helpful comments provided for our paper. We have studied the valuable comments from you and tried our best to revise the manuscript. The revised version has been uploaded, which we would like to submit for your kind consideration.

 Comment 1

Authors should clarify the significance of R values in table 5.

Reply 1

We added the significance of R values in Table 5 and removed Table 5 to Supplemental file 5. Thanks for your suggestions.

 Comment 2

Authors should recheck the references- 7, 11, 18.

Reply 2

Reference 7 and 11 were deleted and reference 18 was replaced with a more appropriate article. Thanks for your suggestions.

Comment 3

It’s interesting that authors ahs mentioned the limitations of this study. Authors should mention how other researchers can learn from these limitations.

Reply 3

We made some modifications in this section in Line 412-420: “Therefore, the dynamic monitor of clinical manifestation and microbial tests should be considered in future researches.

The major superiority of our study was the prospective multi-center design that guaranteed the diversity of patients and reduced the possibility of data bias. Moreover, we included the plasma levels of pro-inflammatory factors into our research and explored the relationship between the reads of mNGS test and pro-inflammatory factors which was rarely studied in previous researches.”

 Thanks for your suggestions.

Comment 4

There are multiple studies done for sepsis and anti-microbial therapy. Authors should mention and discuss them in brief in the introduction section.

Reply 4

We have added some studies of mNGS tests for guidance of sepsis and anti-microbial therapy in the Introduction section. “Besides, the diagnostic efficacy of mNGS won’t be impaired by the application of antibiotics [9]. Some studies also demonstrated that the NGS test could detect antimicrobial resistance (AMR) gene [10,11], and may help clinicians to adjust antibiotic regimen. Apart from bloods samples, mNGS test is applicable to multiple kinds of samples, including resected heart valve [12], cerebrospinal fluid [13], vitreous samples [14], formalin-fixed and paraffin-embedded (FFPE) tissue specimens [15]” (Line 123-129). Besides, we supplemented the experience and superiority of our study in the Discussion section. Thanks again for your kind and helpful advice.

Reviewer 2 Report

The idea of using deep sequencing to identify bacterial infection and antibiotic resistance profile commences over 10 years. This is the only multi-centre investigation on sepsis. However, the difficulty is how to enrich DNA from pathogens from overwhelming host DNA’s . In this manuscript, the authors have not justified why using 300ul plasma, any enrichment for the bacterial/fungi DNAs for sequencing, and the effects of host DNAs, as well as how the number of reads counted and normalized. What are the specific sequences for each strain identified.

Blood culture for bacteria is expected less sensitive than deep sequencing because the culture plates are only suitable for certain bacteria to grow. Fungi appearance may be due to the fungi infection, or long-time use of antibiotic so need to be justified. However, it is useful to do both mNGS and blood culture clinically in near future.  It is an improvement to adjust antibiotics based on the strains identified, but ideally to find the antibiotic resistance elements to guide the usage of antibiotics.

Using mNGS for prognosis is too reluctant but using the number of reads to estimate the bacterial load may be justifiable. The association of mNGS to cytokines is also not justifiable. It would be better to see if any specific cytokine profile is for specific pathogens. Table 3-5 can be deleted and Figure 2 needs to be simplified.

The manuscript writing needs significant improvement too.

Author Response

Reply to Reviewer 2

We are very grateful to your detailed comments for the manuscript. We have amended the relevant part in our manuscript according with your advice.

Comment 1

The idea of using deep sequencing to identify bacterial infection and antibiotic resistance profile commences over 10 years. This is the only multi-centre investigation on sepsis. However, the difficulty is how to enrich DNA from pathogens from overwhelming host DNA’s. In this manuscript, the authors have not justified why using 300ul plasma, any enrichment for the bacterial/fungi DNAs for sequencing, and the effects of host DNAs, as well as how the number of reads counted and normalized. What are the specific sequences for each strain identified.

Reply 1

Thank you for this advice. We have replenished the details about mNGS test in Supplemental file 2. The plasma samples were used to detect suspected pathogens causing sepsis, based on the theory that the nucleic acid of the pathogen will be released into the patient’s plasma after infection. Therefore, the free nucleic acid in plasma could be used to detect bacteria/fungi. About 300μl of plasma was enough for extracting adequate DNA and ensuring the test success. In our test, no enrichment was used to enrich the DNA of bacterial/fungi. The sensitivity of bacterial/fungal detection was ensured by large volume of sequencing data (20M). Besides, the number of strictly aligned sequences of target microorganism detected at the genus/species level were recorded as reads, and no normalization was used when we analyzed the correlation between reads of pathogens and cytokines.

 Comment 2

Blood culture for bacteria is expected less sensitive than deep sequencing because the culture plates are only suitable for certain bacteria to grow. Fungi appearance may be due to the fungi infection, or long-time use of antibiotic so need to be justified. However, it is useful to do both mNGS and blood culture clinically in near future.  It is an improvement to adjust antibiotics based on the strains identified, but ideally to find the antibiotic resistance elements to guide the usage of antibiotics.

Reply 2

We did not detect the antibiotic resistance genes in this study, due to the technical and funding restrictions. Subsequent clinical studies will be conducted and we will make efforts to enroll this part in. We acknowledge this valuable suggestion that providing a reliable and promising research direction.

 Comment 3

Using mNGS for prognosis is too reluctant but using the number of reads to estimate the bacterial load may be justifiable. The association of mNGS to cytokines is also not justifiable. It would be better to see if any specific cytokine profile is for specific pathogens. Table 3-5 can be deleted and Figure 2 needs to be simplified.

Reply 3

We analyzed the correlation between specific cytokine profile and the top three common pathogens, including Klebsiella pneumoniae, Acinetobacter baumannii and Pseudomonas aeruginosa. However, we observed no statistical significance. For interleukin 1-β (IL-1β), eotaxin, interferon-α (IFN-α) and tumor necrosis factor-α (TNF-α), the P value were 0.998, 0.796, 0.577 and 0.620. Therefore, we left out this part in the Results section. Besides, we replaced Table 3-5 to Supplemental files as your suggested. Figure 2 has been simplified and we want it to show the information more clearly.

 Comment 4

The manuscript writing needs significant improvement too.

Reply 4

We carefully proof-read the manuscript to minimize typographical and grammatical errors and did our best to make modification. We sincerely hope that the revised manuscript would meet your requirements.

Reviewer 3 Report

This is an interesting report dealing with septic and non-septic patients in which the metagenomic next generation sequencing (mNGS) has been used in hospital patients with suspected sepsis.  In most cases, bacterial and/or fungal presence has been demonstrated in 277 patients studied, 162 who were diagnosed as being septic.  mNGS seemed faster and more reliable than the traditional culture methods used in hospitals.  There are some issues that need clarification:

·         While the clinical data for septic patients with bacteremia and the mNGS strategy seems sensitive and reliable, there is lack of detail regarding septic patients with fungal infections.  Figure 1 indicates that bacterial infections versus fungal infections had an incidence of 2 and 1.07%, respectively. 

·         There are other data (in red areas and blue areas of Figure 1) that are confusing.  In general, the lack of detail related to frequency of bacterial and fungal infections in septic patients makes it very difficult to determine the value of in vitro measurements related to mNGS, especially in septic patients with fungal infections. 

Unless these confusing data can be revised with additional information, the report will raise a variety of concerns that would undermine the value of the report.

Author Response

Reply to Reviewer 3

Comment:

This is an interesting report dealing with septic and non-septic patients in which the metagenomic next generation sequencing (mNGS) has been used in hospital patients with suspected sepsis.  In most cases, bacterial and/or fungal presence has been demonstrated in 277 patients studied, 162 who were diagnosed as being septic.  mNGS seemed faster and more reliable than the traditional culture methods used in hospitals.  There are some issues that need clarification:

Reply:

Thank you very much for your evaluation and comments on our paper. We have revised the manuscript according to your kind advices.

Comment 1:

While the clinical data for septic patients with bacteremia and the mNGS strategy seems sensitive and reliable, there is lack of detail regarding septic patients with fungal infections.  Figure 1 indicates that bacterial infections versus fungal infections had an incidence of 2 and 1.07%, respectively. 

There are other data (in red areas and blue areas of Figure 1) that are confusing.  In general, the lack of detail related to frequency of bacterial and fungal infections in septic patients makes it very difficult to determine the value of in vitro measurements related to mNGS, especially in septic patients with fungal infections. 

Unless these confusing data can be revised with additional information, the report will raise a variety of concerns that would undermine the value of the report.

Reply 1:

Thanks for your valuable advice. Details about the frequency of bacterial and fungal infections in septic patients were added in the Result section. The overall incidences of bacterial and fungal infections were 42.60% (118/277, red line in Figure 1) and 7.94% (22/277, blue line in Figure 2), respectively. The incidences of sepsis in bacterial and fungal infected patients were 63.56% (75/118) and 54.55% (12/22), respectively. Besides, we calculated the positive rates of mNGS test in bacterial infected and fungi infected patients. The underlying conditions in patients with fungal infections were also briefly described.

“For patients with bacterial infection, 75 were diagnosed as sepsis and the positive rates of blood culture and mNGS test were 36.0% (27/75) and 86.67% (65/75), respectively. Forty-three of them were diagnosed as non-sepsis, while the positive rates of blood culture and mNGS test were 39.53% (17/43) and 90.70% (39/43), respectively. The overall positive rates of blood culture and mNGS test in bacterial-infected patients were 37.29% and 88.14%, respectively.” (Line 254-260)

“For patients with fungi infections (22 cases), 11 of them had histories of organ transplant or malignant tumors, the rest were suffering from long-term pneumonia or severe cardiac dysfunction. Suspected pathogens were Pneumocystis (13 cases), Aspergillus (7 cases), and Candida (2 cases). All of them were infected in lungs. Twelve of fungi infected patients were diagnosed as sepsis, while the rest 10 were non-sepsis. The positive rates of mNGS test were both 100.0% in fungal infections, while only 1 of the 22 patients had positive blood culture result of Candida.” (Line 265-271)

The revised manuscript online has been uploaded, and we sincerely hope this manuscript will meet your requirements. Thank you very much for all your help and best wishes.

Reviewer 4 Report

The authors present a prospective cohort to explore the potential clinical impact of next-generation sequencing for pathogen detection in the treatment of hospitalized patients with suspecte sepsis.

Though this is an interesting topic, the manuscript needs improvement in a number of areas as listed below:

MATERIALS & METHODS:

- It is unclear how the patients were selected for criteron 3 of the inclusion criteria.  For example, if you are testing for bloodstream infection, how were patients enrolled based on a diagnosis of bloodstream infection?  Did they have previous positive cultures, or are the criteria more appropriately stated as 'suspected' infection?

Line 135:  It seems that these blood cultures were obtained at the time of enrollment, and the time of enrollment may have been well after treatment was initiated (as noted in figure 1).  In these cases, were blood cultures available earlier in the course of hospitlization?  If so, what is the result of the cultures obtained before antibiotics and how does this compare?  Also, requiring both cultures to be positive is a fairly high threshold for many organisms, including S aureus and Gram negatives.

Line 167:  There is discussion of Receiver Operating Characteristic (ROC) curve analysis, but it is not shown.  The authors should either include data in the results or delete this text.

RESULTS:

Line 185:  This is not how the Sepsis-3 criteria propose to use the qSOFA.  Rather, the qSOFA is considered to be a screening tool for those with infection to identify the subgroup at high risk for adverse outcomes.

Line 220:  Sentence beginning 'Therefore, for patients ...' should be moved to the discussion.  Also, providing data to support such a statement is necessary.  For example, did patients with fungi identified by NGS but not treated have poor outcomes?  

Line 231:  This was not designed as a noninferiority study and there is no statistical analysis to support such.

Line 246:  Sentence beginning "It is suggested ..." should be moved to the discussion.

There are many other instances of interpretative language within the results section that should be moved to the discussion.

DISCUSSION:

Line 278:  There are many factors that influence modification of antibiotic regimens, not only pathogen identification.  To draw conclusions as stated here without substantial supportive data is somewhat premature.  This is again stated in the paragraph beginning line 319, and could be consolidated.

Line 301:  This is more of a concern with the overall interpretation of the research:  How can you be sure that the pathogens detected by mNGS are not false positives?  What is the reference standard used in determination?  A test cannot serve as its own reference, but it is unclear what that reference might be in this investigation.

Table 2 includes altered mentation in the sepsis / nonsepsis groups, but as noted above, it is one a component of the criteria used to denote sepsis, based on qSOFA.  The other criteria used (qSOFA) should be included as well.

Figure 2 is cluttered and very difficult to interpret.  Are those traditional boxplots?

Author Response

Reply to reviewer 4

We appreciate your extremely constructive and helpful comments provided for our paper. We have studied the valuable comments from you and tried our best to revise the manuscript. The revised version has been uploaded, which we would like to submit for your kind consideration.

 Comment 1

MATERIALS & METHODS:

- It is unclear how the patients were selected for criterion 3 of the inclusion criteria.  For example, if you are testing for bloodstream infection, how were patients enrolled based on a diagnosis of bloodstream infection?  Did they have previous positive cultures, or are the criteria more appropriately stated as 'suspected' infection?

Reply 1

We corrected the 3rd inclusion criteria in Line 150. The diagnosis of bloodstream infection was mainly decided by physicians and principal investigator in each hospital. Most of them didn’t have positive blood cultures. Therefore, it was better to replace “diagnosed” with “suspected”. Thanks for your suggestions.

 Comment 2

MATERIALS & METHODS:

Line 135:  It seems that these blood cultures were obtained at the time of enrollment, and the time of enrollment may have been well after treatment was initiated (as noted in figure 1).  In these cases, were blood cultures available earlier in the course of hospitlization?  If so, what is the result of the cultures obtained before antibiotics and how does this compare?  Also, requiring both cultures to be positive is a fairly high threshold for many organisms, including S aureus and Gram negatives.

Reply 2

There were 31 patients had blood cultures available before the enrollment. All of them conducted blood cultures again after enrollment, and the culture results remained the same. Therefore, no extra analysis was taken towards the blood culture before the enrollment.

As for taking bilateral positivity as a criterion, it was decided by laboratory in each hospital.

Thanks for your suggestions.

 Comment 3

MATERIALS & METHODS:

Line 167:  There is discussion of Receiver Operating Characteristic (ROC) curve analysis, but it is not shown.  The authors should either include data in the results or delete this text.

Reply 3

Receiver Operating Characteristic (ROC) curves were drawn to calculate the value of area under curve (AUC). The AUC values of blood culture and mNGS test were calculated and listed in Line 284. Thanks for your suggestions.

Comment 4

RESULTS:

Line 185:  This is not how the Sepsis-3 criteria propose to use the qSOFA.  Rather, the qSOFA is considered to be a screening tool for those with infection to identify the subgroup at high risk for adverse outcomes.

Reply 4

We admit that our application of the Sepsis-3 criteria needed to be optimized and we will improve this part in future study. Thanks for your suggestions.

Comment 5

RESULTS:

Line 220:  Sentence beginning 'Therefore, for patients ...' should be moved to the discussion.  Also, providing data to support such a statement is necessary.  For example, did patients with fungi identified by NGS but not treated have poor outcomes?

Reply 5

We removed the sentence to the discussion and deleted the inappropriate contents without sufficient arguments. Due to the small number of patients with fungal infections in our study, we did not analyse their prognosis. However, the mNGS was able to detect fungi infection with high sensitivity, future studies will be conducted to explore its prognostic effects. Thanks for your suggestions.

Comment 6

RESULTS:

Line 231:  This was not designed as a noninferiority study and there is no statistical analysis to support such.

Reply 6

We deleted the inappropriate contents without sufficient arguments. Thanks for your suggestions.

Comment 7

RESULTS:

Line 246:  Sentence beginning "It is suggested ..." should be moved to the discussion.

There are many other instances of interpretative language within the results section that should be moved to the discussion.

Reply 7

Sentences have been removed to the discussion section. Thanks for your suggestions.

Comment 8

DISCUSSION:

Line 278:  There are many factors that influence modification of antibiotic regimens, not only pathogen identification.  To draw conclusions as stated here without substantial supportive data is somewhat premature.  This is again stated in the paragraph beginning line 319, and could be consolidated.

Reply 8

We admit that the factors that influence modification of antibiotic regimens are complicated. Therefore, we limited the timing of antibiotic modifications to improve the precision of results. The change of antibiotic agents within 2-7 days after the enrollment, cause the results of mNGS test and blood culture were given during this period. Thanks for your suggestions.

Comment 9

DISCUSSION:

Line 301:  This is more of a concern with the overall interpretation of the research:  How can you be sure that the pathogens detected by mNGS are not false positives?  What is the reference standard used in determination?  A test cannot serve as its own reference, but it is unclear what that reference might be in this investigation.

Reply 9

The positive criteria for mNGS test were list in method section in Line 184-195. As for the last etiological diagnosis, it was made by attending physicians and principal investigator in each hospital, based on microbiological results, clinical features and response to the treatment. Controversial cases were discussed in a seminar of all investigators from different centers and a final diagnosis was determined. If a pathogen was positive in mNGS test while the physicians denied that it caused the infection, then the mNGS test result would be taken as false positive. Thanks for your suggestions.

Comment 10

DISCUSSION:

Table 2 includes altered mentation in the sepsis / nonsepsis groups, but as noted above, it is one a component of the criteria used to denote sepsis, based on qSOFA. The other criteria used (qSOFA) should be included as well.

Reply 10

The patient's respiratory rate and systolic blood pressure were measured multiple times, the data fluctuated rather than fixed. It’s difficult to decide which precise number to take, so they were not included in the analysis. Thanks for your suggestions.

Comment 11

DISCUSSION:

Figure 2 is cluttered and very difficult to interpret.  Are those traditional boxplots?

Reply 11

Figure 2 has been simplified and those were traditional boxplots. Thanks for your suggestions.

Round 2

Reviewer 2 Report

The authors have answered most of question and significantly modified the contents and discussion.

Reviewer 3 Report

Ad requested, the authors have provided detailed information on detection of mNGS along with some new references.  Also some additional clinical information is now provided (Tables 1,2,3).  Finally, extensive clinical details have been removed.  As revised, this report is now much improved.